# Effects of an Easily Implemented Physical Exercise Program on the Ventricular Ejection Fraction of Women with Breast Cancer Undergoing Chemotherapy

**DOI:** 10.3390/ijerph22010094

**Published:** 2025-01-12

**Authors:** Tiago Rafael Moreira, Nuno Pimenta, Alexandra Teixeira, Bruno Nobre Pinheiro, Rui Canário-Lemos, Rafael Peixoto, Nuno Domingos Garrido, José Vilaça-Alves

**Affiliations:** 1Polytechnic Institute of Maia, 4475-690 Maia, Portugal; 2Research Center of Polytechnic Institute of Maia (N2i), 4475-690 Maia, Portugal; 3Local Health Unit of Alto Ave, 4835-044 Guimarães, Portugal; alexandramendesteixeira@hospitaldeguimaraes.min-saude.pt; 4Department of Physical Education, University Unifametro, Fortaleza CE-065, Brazil; bruno.pinheiro@professor.unifametro.edu.br; 5Department of Sport, Exercise and Health Sciences, University of Trás-os-Montes and Alto Douro, 5000-801 Vila Real, Portugal; ruimaldini27@hotmail.com (R.C.-L.); peixoto347@gmail.com (R.P.); ndgarrido@utad.pt (N.D.G.); josevilaca@utad.pt (J.V.-A.); 6Research Center in Sports Sciences, Health Sciences and Human Development, CIDESD, 5000-801 Vila Real, Portugal

**Keywords:** breast cancer, cardiac toxicity, LVEF, chemotherapy

## Abstract

Breast cancer (BC) is the most common cancer among women, with an incidence of 85–94 per 100,000 people annually in Europe. Despite the increasing incidence of BC, advancements in early detection and novel therapeutic approaches have improved survival rates. However, adjuvant treatments are associated with side effects, including a reduction in the left ventricular ejection fraction (LVEF), which can result in severe cardiac damage and progress to heart failure. Methods: Thirty-eight women undergoing BC treatment were included in the study. Eighteen women (47.2 ± 5.4 years old) participated in a supervised physical exercise (PE) program for 60 min/day, twice weekly, at moderate to high intensity (5 min joint mobilization, 20 min predominantly aerobic training, 30 min of strength training, and 5 min cooldown). The remaining 20 women (51.5 ± 9.5 years) were advised to remain active during treatments, but without specific supervision. In the PE group, a slight reduction in the LVEF was observed after chemotherapy (63.73 ± 3.34% vs. 61.00 ± 6.54%, *p* = 0.131). In contrast, the control group showed a statistically significant reduction in the LVEF (64.93 ± 4.00% vs. 60.57 ± 4.86%, *p* = 0.008). Although the results suggest a potential protective effect of regular physical exercise during BC treatment, the study was inconclusive regarding its role in preventing cardiac dysfunction. Further research with a larger sample size and longer follow-up is warranted.

## 1. Introduction

Breast cancer (BC) is the most common cancer among women, with an incidence of 85–94 per 100,000 people annually in Europe [1], and this number is expected to rise. Despite the increasing incidence, the survival rate has significantly improved, with 5-year survival attributed to advancements in early detection and scientific progress [1].

Regular physical exercise (PE) plays a crucial role not only in the prevention of several diseases but also in their management and progression [2,3]. In BC, as in other chronic diseases, PE has demonstrated positive effects on recovery following treatments, including improvements in self-reported quality of life [4], physical fitness [5], and maximal oxygen consumption [6].

BC treatments significantly impact patients’ quality of life, particularly during chemotherapy. Side effects are numerous and include fatigue, cognitive dysfunction, loss of functionality, hair loss, depression, isolation, diarrhea, dyspnea, insomnia, peripheral neuropathy [7], and heart failure due to notable cardiotoxicity [8].

In populations without BC, regular PE has been shown to improve cardiorespiratory fitness and prevent heart failure, among other health benefits [9]. However, despite its well-documented advantages, barriers to PE implementation persist in BC patients, including the physical and emotional burden of treatments and associated symptoms [10].

Few studies have evaluated the impact of PE on the left ventricular ejection fraction (LVEF) in women undergoing chemotherapy. Hojan, K., et al. (2020) [11] reported that PE could mitigate LVEF decline in women receiving trastuzumab therapy. Conversely, in women undergoing anthracycline-based chemotherapy, PE did not yield significant effects on the LVEF [12].

Given that chemotherapy-related cardiotoxicity is one contributor to heart failure, this study aims to assess the effect of a feasible, structured physical exercise program on the LVEF in women undergoing chemotherapy for BC.

## 2. Materials and Methods

The intervention program was approved by the ethics committee of the Hospital Senhora de Oliveira, Guimarães, Portugal (under Ref. 67/2017 CAC). A total of 38 women with BC undergoing chemotherapy treatment were recruited from the same hospital and allocated into two groups for convenience. All participants provided written informed consent before study participation. Eighteen women were included in the experimental group (EG) (14 receiving adjuvant treatment and 4 neoadjuvant treatment), due to their proximity to the hospital, while twenty women were allocated to the control group (CG) (12 receiving adjuvant treatment and 8 neoadjuvant treatment), because their place of residence was farther from the hospital. Inclusion criteria required participants to be over 18 years old, have carcinoma stages I to III, be at the start of chemotherapy treatment, and have no metabolic diseases and/or joint problems preventing physical exercise. The chemotherapy protocols are detailed in Table 1.

Participation approval was obtained from each participant’s oncologist.

### 2.1. Measurements

Anthropometric data, including height and body mass, were collected using a stadiometer (Seca 213), and a calibrated scale (Seca Gmb & Co, Hamburg, Germany, Model: 7621019009), respectively.

The LVEF was evaluated before and after chemotherapy using 3D echocardiography with Doppler (General Eletric Vivid E95^®^, GE Vingmed Ultrasound AS, Horten, Norway) performed by specialized personnel at the Hospital Senhora da Oliveira, with the patient placed in the left lateral decubitus position. LVEF images were acquired from the apical window during a 6- to 8-second inspiratory breath-hold.

### 2.2. Exercise Intervention

The exercise intervention took place during BC treatments over a period of 15.6 ± 4.9 weeks. Participants in the EG performed supervised physical exercise sessions twice a week, each lasting 60 min. The sessions comprised five minutes of joint mobility exercises, twenty minutes of predominantly aerobic training exercises (AT), thirty minutes of strength training exercises, and five minutes of cooldown activities. The AT included step and touch, double lateral displacements, knee and hip flexions, A-steps, and walking/running, performed at moderate intensity (6–7 on the modified Borg scale). The ST (Figure 1) focused on isometric exercises for the upper limbs and isotonic exercises for the trunk and lower limbs. Specific exercises included dynamic squats, isometric horizontal shoulder adductions (using a chair), isometric shoulder flexions (in pairs), isometric horizontal abductions (in pairs), spine flexions (supine position), and isometric hip extensions (supine position). Each exercise was performed for 3 sets of 30 s, with 1 min rest intervals between sets. For isometric exercises, participants were instructed to exert high intensity (8 on the modified Borg scale), while for isotonic exercises, a self-paced cadence was encouraged to ensure safety. All sessions were supervised by an exercise professional. Participants in the CG were advised to maintain physical activity without specific guidelines or supervision.

### 2.3. Statistical Analysis

Data analysis was conducted using the statistical analysis software SPSS (Statistical Package for the Social Sciences, SPSS Science, Chicago, IL, USA), version 21, for Macintosh. Descriptive statistics were performed to characterize variables in terms of central tendency (mean) and dispersion (standard deviation). A graphic analysis was conducted to identify outliers and possible data entry errors. Normality of data distribution was assessed using the Shapiro–Wilk test. For inferential analysis, paired *t*-tests were used to detect significant differences within groups (pre- vs. post-intervention), while independent *t*-tests were employed to compare the EG and CG at each time. Effect sizes were calculated using Cohen’s d, were values of 0.2, 0.5, and 0.8 reflected small, medium, and large effects, respectively. Statistical significance was set at *p* < 0.05.

## 3. Results

The study included 38 women undergoing chemotherapy treatment for BC, with 18 participants allocated to the EG and 20 to the CG. The sample characteristics are presented in Table 2.

In the EG, there were no statistically significant changes in the LVEF values between pre- and post-chemotherapy moments (63.73 ± 3.34 vs. 61.00 ± 6.54; *p* = 0.131; 95%CI = −0.966 to 6.420, d = 0.525). In contrast, the CG showed a statistically significant reduction in the LVEF values after chemotherapy compared to pre-chemotherapy levels (64.93 ± 4.00% vs. 60.57 ± 4.86, *p* = 0.008, 95% CI = 1.356 to 7.358; d = 0.979) (Figure 2). No statistically significant differences between groups were observed at any time point.

## 4. Discussion

The objective of the present study was to analyze the effects of a simple, easily implemented PE program on the LVEF in women with BC undergoing chemotherapy.

The results demonstrated that in the EG there was no severe decline in LVEF values during chemotherapy. In contrast, the CG, which was only advised to maintain regular physical activity, exhibited a statistically significant decrease in LVEF values after chemotherapy compared to pre-treatment levels. Despite this reduction, it was not clinically significant, as no participants developed symptoms of heart failure. The observed LVEF reduction in the CG was approximately 5%, consistent with values reported in other studies examining usual care without structured PE interventions [13]. Additionally, no statistically significant differences were found between EG and CG groups at any time point, although the effect size between groups post-intervention was large. Since LVEF loss caused by chemotherapy is independent of baseline values [14], baseline variations among participants were not considered relevant.

These findings contrast with those of Howden et al., (2019), who analyzed the cardiotoxic effects of trastuzumab in BC patients. Their intervention included 30 min of AT and 30 min of ST, supervised twice weekly, alongside one unsupervised AT session at home. Using a 2:1 periodization paradigm, where exercise intensity was reduced by 10% during treatment weeks, they reported reductions in the LVEF for both the CG (62.8 ± 4.9% vs. 59.1 ± 4.1%) and the EG (64.1 ± 5.0% vs. 60.6 ± 5.4%), with no statistically significant differences between groups [15]. In contrast, the current study applied a flexible undulatory periodization with subjective intensity control. This allowed adjustments in absolute intensity during sessions where participants felt less capable due to chemotherapy, while maintaining relative intensity. Furthermore, as participants in the present study had different cancer subtypes and were not uniformly treated with trastuzumab, this may explain the absence of significant LVEF reductions in the EG, compared to previous studies [13].

Another study involving female BC survivors with cardiomyopathy implemented 16 weeks of AT at 50% heart rate reserve, three times per week for 30 min [16]. The results showed significant improvements in maximum oxygen consumption in the EG compared to the CG (*p* = 0.042). However, no significant changes in the LVEF were observed within or between groups, although the EG increased their LVEF values by 3%. The pre-existing cardiomyopathy in these participants may explain the limited changes, highlighting the importance of initiating PE during chemotherapy to prevent cardiovascular complications.

Few studies have explored the effects of PE during chemotherapy as a preventive measure for LVEF loss and heart failure. Post-treatment studies have reported similar findings to the present study [11]. For instance, women who completed treatment 3 to 6 months prior to intervention exhibited a 1% LVEF loss in the EG and 4% loss in the CG after 9 weeks. Significant differences were observed within the CG and between groups at the post-intervention. The GE intervention included five weekly sessions of combined moderate-intensity training, using treadmills, bicycles, and resistance training equipment. Similarly, other authors found no significant differences in the LVEF between exercise and usual care groups [12].

Current recommendations to mitigate chemotherapy-induced cardiotoxicity focus on pharmacological strategies, such as statins to reduce oxidative and inflammatory effects of anthracyclines, endothelin-1 receptor antagonists to lower inflammatory markers, and stem cell therapies to promote cardiovascular repairs. However, the present study aimed to evaluate the cardioprotective potential of PE in women exposed to chemotherapy.

It is well established that regular PE reduces cardiovascular mortality [17]. In cancer patients, physically active women have a 25% lower risk of cardiovascular death compared to sedentary women [18]. For this reason, the present study sought to implement an accessible exercise program for women undergoing chemotherapy.

Furthermore, muscle contraction during exercise promotes the release of chemokines, such as IL-6, IL-8, IL-15, musclin, apelin, myostatin, and irisin, which help combat treatment-related toxicity and systemic inflammation [19]. This reduction in systemic inflammation may partly explain the prevention of LVEF decline in the EG.

A strength of this study is the demonstration that a simple easily implemented PE program in a hospital setting can promote cardiac health in women undergoing chemotherapy. However, the study has limitations, including the small sample size and the use of 3D echocardiography as the sole method for cardiac function evaluation, and the lack of registration of this study as a clinical trial.

For future studies, it is recommended that LVEF assessment be combined with biochemical markers in a larger sample to validate and expand upon the findings of the present study.

## 5. Conclusions

This study demonstrates that the implementation of an accessible and supervised PE program during chemotherapy may help to mitigate LVEF reduction in women with BC. The findings underscore the importance of integrating exercise into standard oncological care to promote cardiovascular health and potentially reduce mortality risks associated with chemotherapy-induced cardiotoxicity. However, the study has limitations, including a small sample size and the use of 3D echocardiography as the sole method for cardiac function assessment. Future research should address these limitations by including larger cohorts and incorporating biochemical markers to provide a more comprehensive understanding of the cardioprotective mechanisms of PE during cancer treatment.

## Figures and Tables

**Figure 1 ijerph-22-00094-f001:**
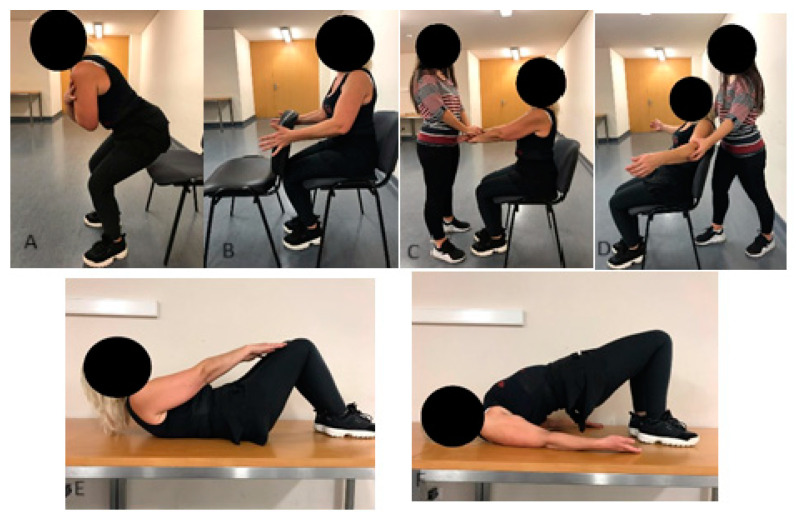
Strength training: (**A**) dynamic squat; (**B**) horizontal shoulder adduction; (**C**) shoulder flexion; (**D**) horizontal shoulder abduction; (**E**) spine flexion; (**F**) hip extension.

**Figure 2 ijerph-22-00094-f002:**
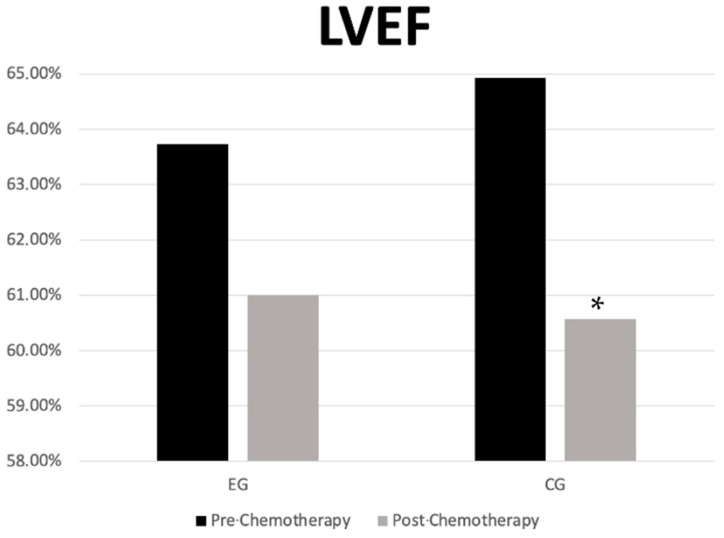
Left ventricular ejection fraction in the experimental and control groups in the pre- and post-chemotherapy moments. * *p* = 0.001 between pre and post. LVEF—left ventricular ejection Fraction; CG—control group; EG—experimental group.

**Table 1 ijerph-22-00094-t001:** Treatments made per group.

EG Treatment	N (18)	CG Treatment	N (20)
AC + Paclitaxel	8	AC + Paclitaxel	13
1FEC + Docetaxel	2	TC	3
TC	4	Paclitaxel	3
AC + Docetaxel	1	Carboplatin + Paclitaxel + AC	1
Paclitaxel + Carboplatin + AC	1	Paclitaxel + Trastuzumab	1
AC	1		
Paclitaxel + Carboplatin	1		

AC—Doxorubicin + Cyclophosphamide; FEC—Fluorouracil + Epirubicin + Cyclophosphamide; TC—Docetaxel + Cyclophosphamide; EG—experimental group; CG—control group.

**Table 2 ijerph-22-00094-t002:** Sample.

	Experimental Group (*n* = 18)	Control Group (*n* = 20)
Age (years)	47.2 ± 5.4	51.5 ± 9.5
Height (m)	1.59 ± 0.1	1.60 ± 0.1
BMI (kg/m^2^)	20.0 ± 3.5	20.8 ± 2.0

Average ± standard deviation.

## Data Availability

The data are stored in the repository of the responsible hospital.

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
