# Peer review of "Effects of an Easily Implemented Physical Exercise Program on the Ventricular Ejection Fraction of Women with Breast Cancer Undergoing Chemotherapy"

_ijerph, 2025, doi:10.3390/ijerph22010094_

Round 1
Reviewer 1 Report
Comments and Suggestions for Authors
Interesting research, which studies a very interesting topic in the relationship between physical exercise and health. This point, I think, should be developed further, especially in the introduction and conclusions. Below I leave suggestions for correction.
In Abstract, there are two abbreviations PE and BC, but they have not been mentioned before, this should be corrected.
Introduction, the first paragraph does not indicate what BC corresponds to. Results and evidence from other studies on the subject should be incorporated, this should be in greater detail. Currently, very general information on the subject and findings is presented, even briefly writing about the benefits of physical exercise in the general population. The last paragraph should be placed in Measures.
In Measures, the 3D echocardiogram analysis evaluation should be detailed. Also indicate the consent procedure and the cycle evaluation, for example, duration, time of day, nutrition, hydration, etc. The procedure for distribution of the control and experimental groups was not indicated.
It is important to highlight the time (weeks) in which the women did the physical exercises, this can even be indicated in the objective, conclusions and title.
In the results, the detail of the women by type of cancer should not be included; this is indicated in Measures.
In discussions, emphasis should be placed on the analysis of the frequency, intensity and duration (weeks) of the recommended exercises. In addition, strengths and limitations of the research conducted should be incorporated.
In conclusions, more details should be given about the studies that should be done, what should these investigations study, or what other variables should be included (something similar is at the end of the discussions). Indicate the number of weeks of physical exercises
Author Response
Dear Reviewer,
Thank you for your recommendations. They will undoubtedly make this work even better.
Regarding the point: “In the Abstract, there are two abbreviations, PE and BC, but they have not been mentioned before; this should be corrected,” this has already been addressed.
For the point: “In the Introduction, the first paragraph does not indicate what BC corresponds to. Results and evidence from other studies on the subject should be incorporated. This should be in greater detail. Currently, very general information on the subject and findings is presented, even briefly writing about the benefits of physical exercise in the general population. The last paragraph should be placed in Measures,” I would like to highlight that the Introduction has been improved according to these suggestions.
In the Methodology section: “In Measures, the 3D echocardiogram analysis evaluation should be detailed. Also indicate the consent procedure and the cycle evaluation, for example, duration, time of day, nutrition, hydration, etc. The procedure for the distribution of the control and experimental groups was not indicated,” these aspects have been improved. The sample was divided based on convenience.
Regarding the duration: “It is important to highlight the time (weeks) in which the women performed the physical exercises; this can even be indicated in the objective, conclusions, and title,” the duration of treatments varies depending on the type of cancer and clinical analyses. Therefore, only an average number of weeks can be provided.
On the point: “In the results, the detail of the women by type of cancer should not be included; this is indicated in Measures,” this factor has been adjusted.
As for: “In discussions, emphasis should be placed on the analysis of the frequency, intensity, and duration (weeks) of the recommended exercises. In addition, strengths and limitations of the research conducted should be incorporated,” the Discussion has been improved, and the study's limitations have been included.
In the Conclusion: “In conclusions, more details should be given about the studies that should be done, what these investigations should study, or what other variables should be included (something similar is at the end of the discussions). Indicate the number of weeks of physical exercises,” this section has been improved, and the requested points have been addressed.
Thank you very much for your review.

Reviewer 2 Report
Comments and Suggestions for Authors
The manuscript entitled “Effects of an Easily Implemented Physical Exercise Program on the Ventricular Ejection Fraction of Women with Breast Cancer undergoing Chemotherapy” was evaluated. The idea of the article is interesting and very valid; however, the work presents numerous methodological and organizational limitations that result in the loss of quality of the manuscript and its methodological merit for the statements to be confirmed.
Below are the main observations:
1- Remove the dot at the end of the title
2- The following information appears in the summary: “85-94/100,0002”. I believe that number 2 at the end of the 100,000 must be wrong. This information also appears in the first line of the article’s introduction.
3- In the summary methods, it is presented that the training protocol has a duration of 60 min, however, when adding the times presented, the total time is 50 minutes (5 min of mobilization, 20 min of aerobic, 20 min of strength and 5 min of cool down). Adjust this information.
4- The methods say that the 13 women who were not trained should remain active, what does this mean? Could this not be a bias for your study, in which they could be active in a training model like the one proposed or even more intense?
5- The abstract also contains the numbers “(3) and (4)”. I believe that these should be citations. However, it is not common to see citations in abstracts. Please check the journal’s rules to see if this is allowed. If so, remember that citations (1) and (2) should appear before 3 and 4.
6 - In the summary results, you mention that there was a small reduction in LVEF in trained women, however, you did not present the “p” values, please present them so we can know if there was significance or not.
7- In the first paragraph of the introduction, the acronym “BC” appears twice without the text stating what they mean. Please check and adjust.
8- The 4th and 5th paragraphs of the introduction are very short. The suggestion is to combine them. Furthermore, paragraph 5 does not present any citation from the literature. I believe that they should include the information from which this statement was taken and add more justifications for the low adherence, not only in public spaces, but in other environments as well.
9- At the end of the introduction, the following sentence was added together: “chemotherapy2. Materials and Methods”. Please adjust it. Furthermore, the information in the paragraph does not include a citation of who mentioned this fact. I believe that you should add information about who stated this, as well as include other articles to reinforce this statement. Epidemiological studies and clinical trials can be used.
10- The methodology should explain how the participants were recruited, as well as the choice of women to make up the training or control group. Was it randomized or by convenience?
11- It was not found whether there was a sample calculation for the number of participants included in the study. Was this done?
12- In the inclusion criteria, it was not clear whether the 22 women were physically active or sedentary at the time of recruitment. Did you evaluate this information?
13- Regarding medication, were all 22 patients being treated with the same medications and doses? Could you include this information?
14- Do you have data on the physical capacity of the patients before and after the intervention? Was the echocardiogram performed only on people at rest without stimulation or was it also evaluated with pharmacological stress? Explain the procedure in more detail so that the reader can know whether the LVEF was basal or stimulated and the reason for this choice.
15- Have you registered the study as a clinical trial? This type of study is a clinical trial and requires registration.
16- At the beginning of the results and in table 1 it is mentioned that the study was composed of 28 women, however, in the entire work only 22 are mentioned. I believe that this information should be corrected. As the study is a clinical trial, it should present the CONSORT for clinical trials so that the number of people invited, and the withdrawals/exclusions are clearer.
17- The LVEF results present values of “IC95= 1.356 – 7.358, d= 0.979”, however, as the t-test was performed, the test value and p value must be presented. This information is chi-square.
18- Regarding the figure, I do not understand why it is classified as figure 7. I did not find figures 1 to 6 in the article. Furthermore, I also do not understand why the data in this figure are presented as percentages. Wouldn't it be more appropriate to present them with absolute values so that the standard deviation can be included? Another thing, you say in the study that the training was efficient, but the figure indicates that both groups of patients reduced LVEF after chemotherapy. It should also include the delta variation pre vs post chemotherapy. The marking of the difference between moments should be signaled within the figure to facilitate visualization by the reader.
19- The discussion is very confusing and does not fully portray the study's proposal. This can be seen by the fact that the methodology and presentation of the results are also confusing. The authors mention that the training had effects, however, they conclude that the findings are inconclusive. I believe that the manuscript needs to be extensively revised to present merit for publication.
20- At the end of the discussion, the authors failed to present the limitations of the study. It was also mentioned that it would be interesting for future work to evaluate cytokines; however, basic information about the patients was also not presented, such as physical capacity tests, sociodemographic information and information about the level of physical activity and whether the participants were already performing the proposed activities before starting the training program.
Author Response
Dear Reviewer,
Thank you for your suggestions. They will help improve this work even further.
Regarding your suggestions for improvement, below are the changes made:
- The dot in the title has been removed.
- The information on incidence has been corrected. There was indeed an error.
- Regarding the duration of the intervention, there was an oversight. The duration was 60 minutes. This has been corrected.
- The number of participants referred to was, by mistake, from the pilot tests conducted. The total number of participants is 38 (18 in the EG and 20 in the CG).
- The citations have been corrected.
- The p-value has been included in the abstract.
- The abbreviations have been corrected.
- The introduction has been improved in the text. The total word count limit set by the journal slightly restricts the broad exploration of topics.
- Regarding this point, I believe it was a formatting change. In the original work, it appears corrected. I hope it follows as indicated.
- The recruitment method and convenience allocation have been added.
- The number of participants was not calculated. The maximum number of people possible was recruited at the hospital center. The number is not high due to the difficulty of having chemotherapy patients exercise.
- The women were physically inactive.
- The women were treated with different treatments. This information has been added.
- We only have information on the physical capacity of the intervention group before and after the treatments. LVEF was evaluated at rest. We also have the GLS values; should we add them?
- We do not have a record.
- The number of participants has been corrected.
- The results values have been corrected.
- The figure numbering has been corrected. LVEF is evaluated as a percentage. For this reason, it appears as a percentage. A mark has also been added for evaluating the significant difference in the CG. Perhaps due to text formatting, it was removed from the image.
- The discussion of the results has been improved. We state that there is an advantage to exercising for heart protection. However, in this sample, with no significant differences between groups, we cannot conclude with certainty.
- The limitations of the study have been added.
Thank you again for your valuable feedback.

Reviewer 3 Report
Comments and Suggestions for Authors
1. At the end of the introduction, the authors highlighted that the main goal of this study is to understand the effect of an easily implemented physical exercise program on the left ventricular ejection fraction (LVEF) in women undergoing BC treatments with chemotherapy; The recommendation is that they highlight the novel/new elements of this article and its role in the scientific literature.
2. The recommendation is for the authors to put greater emphasis on the inclusion and exclusion criteria.
3. It is recommended that the authors introduce the sample size calculation in the study or mention its absence in the limitations of the study.
4. Based on that conclusion: Further studies are necessary on this topic, please put much more emphasis on the novelty part of the study.
5. The conclusions should be re-systematized following the title and the results obtained.
Comments on the Quality of English LanguageModerate editing of the English language is needed.
Author Response
Dear Reviewer,
Thank you for your suggestions. They will help make this work even better.
Regarding your suggestions for improvement, below are the changes made:
1 and 2 - The methodologies have been improved.
3 - The sample size was not calculated. The number of women included was the maximum possible at the hospital center.
4 - Research suggestions have been added.
5 - The conclusions have been improved, and the English has been corrected.
Thank you very much for your suggestions.

Round 2
Reviewer 2 Report
Comments and Suggestions for Authors
Dear Authors,
Thank you for following the suggestions outlined in the review. Below are some adjustments that still need to be made.
Abstract
“In the PE group, a slight reduction in the LVEF was observed after chemotherapy (63.73±3.34% vs. 61.00±6.54%, p=0.131).” I believe that this “p” value does not allow us to make this statement. Remember that to be considered a statistical difference the p value must be less than 0.05.
Introduction
“Hojan, K., et al. (2020) reported that PE could mitigate LVEF decline in women receiving trastuzumab therapy.” Please check if this citation format is in accordance with the journal. Shouldn't it be in numbers?
“LVEF” - Put it in full in the introduction
Materials and Methods
“LVEF was evaluated at rest. We also have the GLS values; should we add them?” - Please add all this information.
“We do not have a record.” - Please register the study as a clinical trial.
Results
“In the EG, there were not statistically significant changes in the LVEF values between pre and post chemotherapy moments (63.73±3.34 vs. 61.00±6.54; p=0.131; 95%CI = -0.966 to 6.420, d=0.525).” - Check this information in the results, for it to be considered a statistical difference the p value must be less than 0.05.
Please take this opportunity to check that you have not included this information in the discussion and conclusion incorrectly. Please review the entire study to avoid misunderstandings.
Author Response
Segue a tradução para o inglês:
Dear Reviewer,
Thank you for your recommendations.
Regarding the points raised, I explain them below:
Abstract:
In the EG, there was no statistically significant difference. In this specific case, this is a positive finding as it suggests the possibility of not developing heart disease in the future, unlike the CG.
Introduction:
The citation format will be reviewed and adjusted by the journal.
LVEF has been added to the introduction.
Materials and Methods:
Regarding GLS, after analysis, rewriting the entire article would be required, so it will not be possible to include it.
Regarding registration as a clinical trial, this has been added as a limitation of the study, as suggested by other reviewers.
Results:
In the EG, there was no statistically significant difference. In this specific case, this is a positive finding as it suggests the possibility of not developing heart disease in the future, unlike the CG.
Thank you once again for your observations.
We hope to have addressed everything appropriately.
Best Regards.